# Nutritional Profiles of *Yoom Noon* Rice from Royal Initiative of Southern Thailand: A Comparison of White Rice, Brown Rice, and Germinated Brown Rice

**DOI:** 10.3390/foods12152952

**Published:** 2023-08-04

**Authors:** Pijug Summpunn, Nattharika Deh-ae, Worawan Panpipat, Supranee Manurakchinakorn, Phuangthip Bhoopong, Natthawuddhi Donlao, Saroat Rawdkuen, Kalidas Shetty, Manat Chaijan

**Affiliations:** 1Food Technology and Innovation Research Center of Excellence, School of Agricultural Technology and Food Industry, Walailak University, Nakhon Si Thammarat 80160, Thailand; pijug.su@wu.ac.th (P.S.); nattharika.de@mail.wu.ac.th (N.D.-a.); pworawan@wu.ac.th (W.P.); schinakorn@yahoo.com (S.M.); bphuangt@wu.ac.th (P.B.); 2Food Science and Technology Program, School of Agro-Industry, Mae Fah Luang University, Chiang Rai 57100, Thailand; natthawuddhi.don@mfu.ac.th (N.D.); saroat@mfu.ac.th (S.R.); 3Global Institute of Food Security and International Agriculture (GIFSIA), North Dakota State University, 374 D Loftsgard Hall, 1360 Albrecht Blvd., Fargo, ND 58108, USA; kalidas.shetty@ndsu.edu

**Keywords:** rice, nutrition, food security, food sustainability, germination

## Abstract

For long-term food sustainability and security, it is crucial to recognize and preserve Indigenous rice varieties and their diversity. *Yoom Noon* is one of the non-glutinous rice (*Oryza sativa* L.) varieties being conserved as part of the Phanang Basin Area Development Project, which is administered by the Royal Initiative of Nakhon Si Thammarat in Southern Thailand. The goal of this research was to compare the nutritional profiles of *Yoom Noon* white rice, brown rice, and germinated brown rice. The results indicated that carbohydrate content was found to be the most plentiful macronutrient in all processed *Yoom Noon* rice types, accounting for 67.1 to 81.5% of the total. White rice had the highest carbohydrate content (*p* < 0.05), followed by brown rice and germinated brown rice. Brown rice had more protein and fat than white rice (*p* < 0.05). The maximum protein, dietary fiber, and ash content were found in germinated brown rice, followed by brown rice and white rice (*p* < 0.05). White rice had the highest amylose content, around 24% (*p* < 0.05), followed by brown rice (22%), and germinated brown rice (20%). Mg levels in all white, brown, and germinated brown rice ranged from 6.59 to 10.59 mg/100 g, which was shown to be the highest among the minerals studied (*p* < 0.05). Zn (4.10–6.18 mg/100 g) was the second most abundant mineral, followed by Fe (3.45–4.92 mg/100 g), K (2.61–3.81 mg/100 g), Mn (1.20–4.48 mg/100 g), Ca (1.14–1.66 mg/100 g), and Cu (0.16–0.23 mg/100 g). Se was not found in any processed *Yoom Noon* rice. Overall, brown rice had the highest content of macro- and micronutrients (*p* < 0.05). In all processed rice, thiamin was found in the highest amount (56–85 mg/100 g), followed by pyridoxine (18–44 g/100 g) and nicotinamide (4–45 g/100 g) (*p* < 0.05). Riboflavin was not identified in any of the three types of processed *Yoom Noon* rice. Individual vitamin concentrations varied among processed rice, with germinated brown rice having the highest thiamine content by around 1.5 and 1.3 folds compared to white and brown rice, respectively. The GABA level was the highest in germinated rice (585 mg/kg), which was around three times higher than in brown rice (*p* < 0.05), whereas GABA was not detectable in white rice. The greatest total extractable flavonoid level was found in brown rice (495 mg rutin equivalent (RE)/100 g), followed by germinated brown rice (232 mg RE/100 g), while white rice had no detectable total extractable flavonoid. Brown rice had the highest phytic acid level (11.2 mg/100 g), which was 1.2 times higher than germinated brown rice (*p* < 0.05). However, phytic acid was not detected in white rice. White rice (10.25 mg/100 g) and brown rice (10.04 mg/100 g) had the highest non-significant rapidly available glucose (RAG) values, while germinated brown rice had the lowest (5.33 mg/100 g). In contrast, germinated brown rice had the highest slowly available glucose (SAG) value (9.19 mg/100 g), followed by brown rice (3.58 mg/100 g) and white rice (1.61 mg/100 g) (*p* < 0.05).

## 1. Introduction

Rice (*Oryza sativa* L.) is a good and amenable food delivery system for nutrients for a wide population of humanity due to its high intake and significant supply of carbohydrates, vitamins, minerals, and bioactive compounds [1,2,3,4,5,6]. By 2050, it is predicted that global rice production will treble as a result of growing consumer demand [7].

The nutritional characteristics, as well as other compositional, physicochemical, and bioactive qualities of rice, can be affected by processing techniques such as milling and germination [6,8]. Brown rice is a crucial variety of whole grains that has attracted a lot of interest because of its possible health advantages, and this has recently received a lot of attention [8]. Three layers make up brown rice: the bran, the embryo, and the endosperm; the majority of the bran and embryo are eliminated during the production of white rice [8]. Thus, polyphenolics, dietary fiber, minerals, vitamins, carotenoids, and γ-oryzanols are among the many phytonutrients that are plentiful in brown rice [9,10]. The unwanted degradation of key cellular biomolecules such as proteins, lipids, and DNA may be prevented by these substances acting as natural antioxidants [9,11]. Additionally, the nutritional value of whole grains is significantly increased during germination [4,6]. Germinated rice has recently drawn more attention as a functional food due to its health-promoting benefits and highly nutritious attributes, such as vitamins, fiber, amino acids, minerals, flavonoids, and phenolic acids [4,5,6,12]. The bioprocessing strategy of germinating promotes the formation of secondary metabolites, including polyphenols and γ-aminobutyric acid (GABA) [2,6,13]. As public awareness of nutrition and health has increased, research has recently evolved to focus on the development of functional food products with bioactive components. This is also relevant in rice where numerous studies [1,2,3,6,14] have been conducted to determine how refining and germination affect the functional, physicochemical, and nutritional aspects of different rice cultivars.

Therefore, advancing rice functional foods and ingredients from Indigenous rice, also known as traditional rice, is essential to the needs of health-targeted security and sustainability of the food supply. These traditional rice varieties have been grown by Indigenous people for many generations, who have modified them to fit their traditions and environments. Indigenous rice is deeply ingrained in the cultural traditions and legacies of Indigenous populations. These variations connect ancestry, identity, and knowledge and are significant from a cultural and social standpoint [15,16]. Protecting Indigenous food sources also entails preserving traditional practices and cultural heritage associated with its cultivation, which contributes to the overall diversity and richness of global culinary traditions [17]. Indigenous rice is, therefore, essential to sustaining food security and national sovereignty in an era of rising globalization and population [2,4,5,6]. Communities can lessen their reliance on external food supplies and exert more control over their own food systems by encouraging the growth and consumption of an array of rice varieties [6,18,19].

In Thailand, various native rice cultivars have been grown. In Southern Thailand, more than 4000 regional rice varieties have been found [6]. Domestic non-glutinous rice cultivars have been raised rapidly and consistently using the same farming methods as organic Thai rice, particularly in Nakhon Si Thammarat. However, *Yoom Noon*, one of the native rice varieties, has been planted more frequently as a result of the Royal Initiative to preserve the Indigenous rice varieties [6]. *Yoom Noon* is one of the rice types under conservation in the Phanang Basin Area Development Project under the Royal Initiative of Nakhon Si Thammarat, Southern Thailand [19]. In Southern Thailand, “*Yoom Noon*” refers to “jackfruit”. Local farmers named this rice variety due to the mild jackfruit scent it gives off when cooked [19]. Previously, the antioxidant compositions and in vitro antioxidant functionality of *Yoom Noon* rice processed with different techniques (brown rice, germinated brown rice, white rice, and rice grass) were studied alongside four varieties of Southern Thai rice (*Kaab Dum*, *Khai Mod Rin*, *Look Lai*, and *Yar Ko*) [6]. Rice’s antioxidant compounds and antioxidative capacities were shown to be significantly influenced by both variety and processing. All rice cultivars contained high quantities of total extractable phenolic compounds and carotenoids, particularly rice grass and germinated brown rice [6]. Furthermore, a greater γ-oryzanol content was discovered following germination. After sprouting, all rice cultivars had greater phenolic compounds, ascorbic acid, and carotenoid contents. Overall, the highest total phenolic content was found in *Yoom Noon* rice grass. The greatest γ-oryzanol content was found in *Yoom Noon’s* germinated brown rice. The aqueous extracts of all rice cultivars had exceptional ABTS free radical scavenging activity [6]. The findings could be applied as a guideline to choose the best rice variety and processing method to meet the requirements of farmers who like to advance rice as a functional food ingredient and to encourage health-conscious customers to consume Indigenous rice, with *Yoom Noon* being a good choice as a result of this study. However, it is important to determine the quality of the nutritional value of *Yoom Noon* rice as a fundamental standard for healthy consumption in order to consider its potential applications and to support the concept of healthy food ingredients. Therefore, the goal of this study was to examine the nutritional profiles of the white, brown, and germinated forms of *Yoom Noon* rice from Southern Thailand. Investigating the nutritional benefits of Indigenous rice not only advances scientific understanding but also sets the pathway for supporting sustainable agriculture and cultural preservation.

## 2. Materials and Methods

### 2.1. Chemicals

All chemicals, such as gamma-amino butyric acid (GABA), sodium acetate, and ethanol, were procured from Sigma-Aldrich Corp. (St. Louis, MO, USA). Enzymes like amyloglucosidase, amylase (heat-stable), and pancreatin were also obtained from Sigma-Aldrich.

### 2.2. White Rice, Brown Rice, and Germinated Brown Rice Samples

A native Southern Thai non-glutinous rice (*Oryza sativa* L.), var. *Yoom Noon*, was harvested in Ban Phoeng, Pak Phanang, Nakhon Si Thammarat, Thailand. A total of 300 kg of paddy rice was used. White and brown rice were manufactured using a domestic method, and nutritional contents were compared to germinated brown rice (Figure 1). To acquire brown rice samples with intact bran layers and germ, the paddies were milled to remove the hull using a home-scale miller model (THU35B; SATAKE, Hiroshima, Japan). The bran layer was removed during the next round of milling, yielding white rice samples. The rice seeds were tested for germinability (at least 90% germination) before being used to produce germinated brown rice. To prepare the germinated brown rice, brown rice was soaked in water (pH = 5) with a brown rice-to-water ratio of 1:4 for 96 h at 35 °C, changing the water every 6 h, and then towel dried [6]. To prepare the samples for analysis, white rice, brown rice, and germinated brown rice were ground for 5 min using a grinder (MK 5087M Panasonic Food Processor, Selangor Darul Ehsan, Malaysia). The samples were sealed in ethylene-vinyl alcohol copolymer (EVOH) bags and stored at −20 °C until used. The storage period was no longer than 1 month.

### 2.3. Analyses

#### 2.3.1. Proximate Composition and Amylose

For proximate composition analysis, the AOAC [20] standard methods were used. Moisture, protein (a conversion factor = 5.95), ash, fiber, fat, and carbohydrate (calculated by difference) contents were determined.

Amylose content was evaluated using the method of Chumsri et al. [5]. The sample (100 mg) was mixed with 1 mL of 95% (*v*/*v*) ethanol and 9 mL of 2 M NaOH. The mixture was raised to 100 mL with deionized water (DI), and then 2 mL of 0.2% (*w*/*v*) iodine solution was added. After that, the absorbance was measured at 620 nm with a spectrophotometer (Shimadzu UV-2100, Columbia, MD, USA). A calibration curve made with standard potato amylose was used to calculate the amylose content.

#### 2.3.2. Mineral and Vitamin

Magnesium (Mg), zinc (Zn), iron (Fe), calcium (Ca), potassium (K), manganese (Mn), copper (Cu), and selenium (Se) were measured using the AOAC method [20]. Samples (4 g) were mixed with strong nitric acid (4 mL) and violently shaken for 5 min. On a hot plate, the combinations were heated until digestion was accomplished. The digested products were placed into a volumetric flask and poured to a total volume of 10 mL with DI. An inductively coupled plasma optical emission spectrophotometer (Perkin-Elmer, Model 4300DV, Norwalk, CT, USA) was used to analyze the solution. Flow rates of argon to plasma, auxiliary, and nebulizer were set at 15, 0.2, and 0.8 L/min, respectively. The flow rate of sample was set at 1.5 mL/min.

Water-soluble vitamins such as thiamin (B1), riboflavin (B2), nicotinamide (B3), and pyridoxine (B6) were measured using high-performance liquid chromatography (HPLC) methods [5,21]. A 25 mL solution of 0.1 N H_2_SO_4_ was added to the sample (2 g), which was then incubated at 121 °C for 30 min. After cooling, a 2.5 M sodium acetate was utilized to adjust the pH to 4.5. The mixture was left to incubate overnight at 35 °C after 50 mg of α-amylase was added. The resulting filtrate was mixed with distilled water to 50 mL after filtering with a Whatman No. 4 filter paper before being filtered again with a 0.45 μm Millipore filter. To separate the sample (5 μL injection), a C18-MS II Cosmosil (46 × 150 mm) column, Purosphere (Merck), was employed. As a mobile phase, absolute methanol:25 mM phosphate buffer pH 3.5 (33:67, *v*/*v*) was utilized with an isocratic flow of 0.6 mL/min, 25 °C, and UV scanned at 160–800 nm for detection. The standard solution calibration curves were performed.

#### 2.3.3. GABA, Total Extractable Flavonoid (TEF), and Phytic Acid (PA)

For GABA analysis, the method of Zhang et al. [13] was used. Sample (1 g) was mixed with 5 mL of deionized water (DI). The mixture was oscillated and extracted for 1 h before being filtered (Whatman No. 1). After that, 0.5 mL of the filtrate was collected and mixed with 0.2 mL of 0.2 M borate buffer (pH 9.0), 1 mL of 6% phenol, and 0.4 mL of 9% sodium hypochlorite. After intense oscillation, the combination was immersed in boiling water for 10 min, then transferred to an ice bath for 20 min, and continuously oscillated until a blue color developed. Thereafter, 2 mL of 60% ethanol was added, and the sample was spectrophotometrically examined at 645 nm. Standard curve was prepared using standard GABA.

For TEF, a 0.5 g sample was extracted repeatedly with 8 mL of 80% methanol) at 35 °C for 1 h in an ultrasonic bath (40 kHz, Ultrasons-H model 3000841 JP Selecta, Barcelona, Spain. After centrifugation at 5000× *g* for 25 min (RC-5B plus centrifuge, Sorvall, Norwalk, CT, USA), the free flavonoid fraction was obtained. The pH of the supernatant was then adjusted to 4.5–5.5 using 6 M HCl. The aforementioned residue was washed once again with 20 mL of distilled water. Following water elimination, the samples were ultrasonically blended twice with 4 M NaOH (20 mL) for 2 h. Using 6 M HCl, the pH of the mixture was elevated to 4.5–5.5. Following centrifugation under the same conditions, the bound flavonoid fraction supernatant was recovered [6]. The TEF content of the combined fractions was then determined. In a nutshell, 5 mL of 2% methanolic AlCl_3_ was combined with 5 mL of extract. After a 10 min incubation at 35 °C, the absorbance at 415 nm was measured with a Libra S22 UV-vis spectrophotometer. A control sample with no AlCl_3_ was used. TEF was determined using the rutin standard curve and represented as mg rutin equivalent (RE)/100 g sample [22].

For PA, the sample (0.3 g) was combined with 10 mL of 0.2 M HCl and shaken for 2 h before centrifugation (10,000× *g*/10 min/25 °C). Then, 2.5 mL of supernatant was combined with 0.2% FeCl_3_ (2 mL), which was then heated in a water bath for 30 min before cooling and centrifuging at 10,000× *g* for 15 min. Consequently, the supernatant was discarded, and the remainder was repeatedly rinsed with 5 mL of DI. The residue was treated with 3 mL of 1.5 M NaOH, vortexed for 2 min, and centrifuged (10,000× *g*/10 min). The supernatant was eliminated, and the remaining mixture was dissolved in 10 mL of 0.5 M HCl. To make the final volume of 50 mL, DI was added. An atomic absorption spectrophotometer (Shimadzu, AA6300, Tokyo, Japan) was used to analyze the Fe content of the solution. The PA content was estimated by multiplying using the factor of 4.2 [23].

#### 2.3.4. Rapidly Available Glucose (RAG) and Slowly Available Glucose (SAG)

RAG and SAG of cooked rice were analyzed using the procedure of Englyst et al. [24,25]. Cooked rice was prepared using the domestic rice cooker. Briefly, cooked rice (3 g) was incubated with a mixture of enzymes (amyloglucosidase, amylase, and pancreatin) under controlled conditions (37 °C/pH 5.2). Viscosity was standardized using guar gum in the mixture, according to the Englyst protocol [24]. Subsamples were taken from the mixture at certain time periods (20 and 120 min) and analyzed for glucose content, which was then utilized to determine the RAG and SAG values.
RAG (g/100 g) = (G20 × 100)/w(1)
SAG (g/100 g) = [(G120 − G20) × 100]/w(2)
where w = weight of sample (g).

### 2.4. Statistical Analysis

All of the tests were carried out in triplicate (*n* = 3). On the data, an ANOVA analysis was conducted. Duncan’s multiple range analysis was used to compare the means. The statistical analysis was performed using SPSS 23.0 (SPSS Inc., Chicago, IL, USA).

## 3. Results and Discussion

### 3.1. Proximate Composition and Amylose Content

The processing stages may alter the presence of key nutrients in rice. As a result, the proximate composition of three different processed *Yoom Noon* rice was investigated, as shown in Figure 2. Carbohydrate, primarily starch, was the most abundant macronutrient in all processed *Yoom Noon* rice, accounting for 67.1 to 81.5% of the total. White rice contained the highest amount of carbohydrates, followed by brown rice and germinated brown rice (*p* < 0.05). The processed methods influenced carbohydrate reimaging, with polishing significantly removing the aleurone outermost layer (mixed tissues of the pericarp, testa, aleurone, and embryo) of rice grain and leaving starchy endosperm [26]. Brown rice, on the other hand, had aleurone outermost layers of proteins, minerals, lipids, and fibers, resulting in fewer carbohydrates than white rice. *Yoom Noon* brown rice has a higher protein and lipid content than white rice, which corresponds to its lower carbohydrate counterpart. The same explanation was provided for germinated brown rice, which contained less carbohydrates than white rice. Moisture is the second most common component found in all processed *Yoom Noon* rice, with germinated brown rice having the highest moisture content (19%). This was because water was significantly diffused to the rice grain during the immersion step, which was required for germination and sprouting. Brown rice and white rice had moisture contents of 12% and 10%, respectively, which were within the 11–14% range for standard rice moisture content, allowing for long-term storage [27]. Protein content ranged from 7 to 11%, while lipid content ranged from 0.8 to 1.8% (Figure 2). Germinated brown rice had the greatest protein content (11%), followed by brown rice (9%) and white rice (7%). The increase in protein content caused by germination could be attributed to the production of some proteins, peptides, and amino acids during renewed protein biosynthesis. Protein synthesis is thought to have increased in three stages: (a) concomitant with swelling; (b) during the lag phase between the end of water intake and the start of development; (c) shortly following protrusion through the seed coat [28]. The additional source of nitrogen that serves as a precursor for protein synthesis may come from the air and water that the seed may absorb during soaking and germination. The increased protein content of germinated *Yoom Noon* brown rice may provide a nutritional advantage. Brown rice had the highest lipid content (1.8%), followed by germinated brown rice (1.4%) and white rice (0.8%), respectively (Figure 2). Lipids are typically deposited in rice germ, which is largely removed after milling [29]. Kim et al. [30] found a slight difference in lipid content between non-germinated and germinated brown rice, which was consistent with our findings. Germinated brown rice had a higher dietary fiber content of 2.2% compared to brown rice (1.8%), while white rice possessed the lowest dietary fiber content (0.7%). The dietary fiber content increased after rice germination, most likely due to the synthesis of new constituents upon germination [30]. Furthermore, the increase in brown rice’s dietary fiber content is primarily due to insoluble dietary fiber in aleurone cell walls [31]. The highest ash content was found in germinated brown rice (1.5%), followed by brown rice (0.9%) and white rice (0.7%). The amount of ash in rice is generally related to mineral element quantities [4]. The high concentration of ash in germinated brown rice may result from the adsorption of several minerals required for germination [32]. According to Kranner and Colville [33], efficient mineral retention in seeds is required for the proper germination and development of subsequent vegetative phases of rice crops, such as the high Fe requirement during the rice life cycle [34]. Mineral elements are required for seed germination because nutrients stored in seed structures are mobilized during germination for various growth and metabolic needs [35]. The presence of aleurone layers with high mineral content in brown rice may explain its higher ash content when compared to white rice [36]. According to our previous studies on the proximate composition of two brown Indigenous rice, the moisture content ranged from 5.00 to 5.06%, the protein content ranged from 6.35 to 8.14%, the ash content ranged from 2.0 to 2.2%, the crude fiber content ranged from 2.06 to 2.85%, the fat content ranged from 1.08 to 2.0%, and the carbohydrate content ranged from 85.79 to 87.82% [4]. The difference could be due to the rice cultivar and processing methods. Brown rice is processed minimally and, thus, retains most of the original nutrients within the grain, whereas white rice or polished rice lacks most nutrients because they are forced into the grain’s husk during manufacturing and then eliminated throughout polishing [37]. In addition, the decomposition of high-molecular-weight polymers during germination yields bio-functional substances and improves nutritional and organoleptic properties [38]. From the results, brown rice and germinated brown rice provide more essential nutrients than white rice.

The different processing methods were aligned to the amylose content of *Yoom Noon* rice, as shown in Figure 3. White rice had the highest amylose content, approximately 24%, followed by brown rice (22%) and germinated brown rice (20%), respectively (*p* < 0.05). Sun et al. [39] determined that the starch content, particularly amylose and amylopectin, varies depending on the processing method. Typically, rice with a high amylose content has a lower glycemic index (GI) because amylose is more difficult to metabolize as part of the diet than simple sugars like glucose [40]. As a result, it ensures a sustained release of sugar into the blood without spiking immediately after a meal. The other components were largely removed during the polishing of white rice, resulting in a higher concentration of starch. Similarly, the higher presence of non-starch components in brown rice resulted in a lower amylose content [41]. This was in line with the results of Chatterjee and Das [41], who observed that the amylose content of polished samples ranged from 11.06 to 19.40% in all ten rice varieties due to the fact that starch is mostly concentrated in the endosperm and less in the bran. Amylose-bound complexes with other substances, such as proteins and lipids, might also result in less extractable amylose in the solution leading to a lower relative amylose content measured from brown rice. Likewise, the low amylose content of germinated *Yoom Noon* rice may have contributed to the activation of endogenous amylase, resulting in amylose molecule decomposition. This process could be related to sprouting, which requires the use of stored starch for seedling growth. The findings were consistent with Wu et al. [42], who discovered a decrease in total starch, amylose, and amylopectin content during brown rice germination. Similarly, Zheng et al. [43] observed that the total starch, amylose, and amylopectin contents of brown rice gradually decreased after germination, with amylose content decreasing faster than amylopectin content. Some amylose leaching during brown rice soaking may also be associated with a reduction in amylose in final germinated brown rice (Figure 3). Amylose, due to its lower molecular weight, is more easily converted into a water-soluble component than amylopectin. Yenrina et al. [44] hypothesized that amylose is easier to become a water-soluble component compared to amylopectin due to its smaller molecular weight. According to the results, the amylose content of *Yoom Noon* rice varies depending on the processing methods.

### 3.2. Mineral Profiles

Minerals, both major and minor content, including K, Na, Mg, Ca, Zn, Mn, Fe, Cu, Al, and Cr, are regarded as essential and significant in biological systems because of their potential to regulate bodily functions [45]. The mineral profile of three different processed Indigenous *Yoom Noon* rice is shown in Table 1. Mg content ranged from 6.59 to 10.59 mg/100 g in all white, brown, and germinated brown *Yoom Noon* rice, which was found to be the highest among all investigated (*p* < 0.05). The second highest mineral content in *Yoom Noon* rice was Zn (4.10–6.18 mg/100 g), followed by Fe (3.45–4.92 mg/100 g), K (2.61–3.81 mg/100 g), Mn (1.20–4.48 mg/100 g), Ca (1.14–1.66 mg/100 g), and Cu (0.16–0.23 mg/100 g), respectively (*p* < 0.05, Table 1). It should be noted that Se was not found in any of the three types of processed *Yoom Noon* rice. All minerals reported in this study had a higher content than 10 dehulled rice varieties from Pakistan [45], indicating that Indigenous *Yoom Noon* rice is a rich source of minerals. According to Sangha and Sachdeva [46], different rice genotypes have different mineral concentrations. There were some similarities and differences between our study and Huang et al. [47], who observed that brown rice contains an average amount of minerals, with a clear presence of P (4652 mg/kg) and K (3810 mg/kg), followed by Mg (1558 mg/kg), and a very low amount of Zn (34 mg/kg) and Fe (12 mg/kg). The difference between our study and others could be attributed to mineral content accumulation in grains being significantly affected by planting sites with variations in soil composition [48]. Brown rice had the highest content of macro- and micro-mineral elements among the processed *Yoom Noon* rice types (*p* < 0.05, Table 1). Mineral elements are concentrated primarily in the outer layers of rice grains [49], with rice bran accounting for 61% of the total grain mineral content [50]. The presence of aleurone layers in brown rice is associated with higher levels of Fe and Zn compared to other processed rice [34]. Furthermore, rice polishing reduced Mn and Fe content by 50% and 60%, respectively [29], which explains why white rice contained less of both minerals than brown rice. This was in agreement with Lu et al. [32], who found that the mineral concentration in polished rice (endosperm) was smaller than in hull and bran (embryo + aleurone layer). The relative concentrations of K, Fe, and Zn in the different grain portions were as follows: bran > hull > whole grain > brown rice > polished rice, whereas Ca and Mn were as follows: hull > bran > whole grain > brown rice > polished rice [32]. Except for Mg and Mn, germinated brown rice contained fewer macro- and micro-mineral elements than white rice (*p* < 0.05). This could be attributed to the partial leaching of these minerals during the soaking step of the germination process as well as the utilization of specific minerals during germination. This was in agreement with the findings of Rahman et al. [51], who observed that the longer the sprouting stage or the longer the immersion and germination steps, the lower the Mg and Fe content. The act of immersion is primarily responsible for mineral solubility, allowing minerals in rice to be reduced. Typically, mineral content, solubility, and bioavailability change during germination or pre-germination [52]. Although germinated brown rice had the highest ash content (Figure 2), it appeared to have lower concentrations of the eight tested minerals than that of brown rice, which could be explained by the fact that numerous minerals were untested, such as P and Na.

### 3.3. Vitamin Profiles, GABA, TEF, and PA

Table 2 illustrates the vitamin content of three different processed *Yoom Noon* rice. Thiamin content was the highest (56–85 mg/100 g), followed by pyridoxine (18–44 g/100 g) and nicotinamide (4–45 g/100 g) (*p* < 0.05). Riboflavin was not found in any of the three processed types of *Yoom noon*. It should be noted that thiamin was the most abundant vitamin present in all processed *Yoom noon* rice, outnumbering the other two vitamins tested. Chaudhari et al. [53] suggested that rice is high in B-complex vitamins like thiamin, riboflavin, and niacin, which replenish the skin and blood vessels while also lowering LDL-cholesterol. There was some variation in individual vitamin concentrations among differently processed *Yoom Noon* rice, with germinated brown rice having the highest thiamin content by roughly 1.52 and 1.31 folds compared to white and brown rice, respectively (Table 2). This corresponded to our earlier report [2], which showed that thiamin content increased during the germination of some Thai rice cultivars. Similarly, germinated Korean rice had a 1.70–2.10 folds increase in thiamin content compared to brown rice [54]. Brown rice had the highest nicotinamide content (45 μg/100 g), which was 11.25 times higher than white rice and 1.45 times higher than germinated brown rice (*p* < 0.05). Because *L*-tryptophan is a precursor for nicotinamide synthesis, its presence in rice is strongly related to the amount of nicotinamide [55]. The loss of protein during rice polishing could explain why nicotinamide levels are lower in white *Yoom Noon* rice. Furthermore, the loss of precursor substrate, *L*-tryptophan, or water-soluble nicotinamide during the rice soaking step of germination may result in a decrease in nicotinamide in germinated brown rice. It should be noted that there is a smaller difference in nicotinamide content between brown *Yoom Noon* rice and its germinated counterpart (Table 2). In contrast to our findings, Jeong et al. [54] found an increase in nicotinamide content after the germination of three Korean rice cultivars. The difference could be attributed to the different rice cultivars and germination processes. The same pattern was observed in pyridoxine content, with brown rice having the highest concentration (44 μg/100 g), followed by germinated brown rice (21 μg/100 g) and white rice (18 μg/100 g). This study found that the tested vitamin content of *Yoom Noon* rice varies depending on the processing method used, with white rice having the lowest tested vitamins. The milling and polishing processes destroy 67% of vitamin B3, 80% of vitamin B1, and 90% of vitamin B6 present in the raw unmilled variety [29].

GABA has been suggested to be useful in lowering blood pressure, managing hyperglycemia and cholesterol levels, accelerating brain cell metabolism, preventing cancer and Alzheimer’s disease, and treating anxiety-related conditions [56]. GABA biosynthesis involves the decarboxylation of glutamic acid, which is catalyzed by glutamate decarboxylase (GAD) during seed germination. The highest GABA content was found in germinated *Yoom Noon* rice (585 mg/kg), which was 2.97 times greater than brown rice (*p* < 0.05, Table 2). GABA was not detected in white rice. Chaijan and Panpipat [2] observed that the GABA content of two Thai rice varieties, *Khemtong* and *Khai Mod Rin*, increased during germination. Similarly, the GABA content of rough rice increased from 15.34 mg/100 g to 31.79 mg/100 g after germination [30]. Germination is primarily a metabolic process that elevates the synthesis of secondary metabolites, particularly GABA [57], which is related to substrate level, glutamic acid, and GAD activity [58].

Flavonoids are produced by the phenylpropanoid metabolic pathway, which consists primarily of a 15-carbon skeleton residing in two aromatic rings (A- and B-rings) interconnected by a 3-carbon chain (structure C6-C3-C6) [59]. Flavonoids are recognized for their capability to provide electrons and halt chain reactions, resulting in potent antioxidant activity. Brown *Yoom Noon* rice contained the largest amount of total extractable flavonoid (TEF) (495 mg RE/100 g), followed by germinated rice (232 mg RE/100 g), and white *Yoom Noon* rice had no detectable TEF (Table 2). Because flavonoids are mostly deposited in the bran and outer layer of rice grain, brown rice had a higher concentration of TEF in this study. Goufo and Trindade [59] reported that TFC is typically found in the bran (576.8 mg/100 g), which is higher than the husk (1.8 times), whole grain (3.1–4.2 times), and endosperm (5.4–15.6 times). This also supported our finding that TEF was not detectable in white *Yoom Noon* rice (Table 2). Furthermore, flavonoid loss in germinated *Yoom Noon* rice caused by the leaching effect during the immersion step of germination resulted in lower TEF content in germinated *Yoom Noon* brown rice compared to brown rice (*p* < 0.05). Seven flavonoids are typically reported in rice, with tricin appearing to be the most abundant in the bran, reaching approximately 77%, and others showing up in the following order: luteolin (14%), apigenin (6%), quercetin (3%) iso-rhamnetin (1%), kaempferol (1%), and myricetin (1%) [59]. Considering our results, polishing and germination may have adverse effects on the TEF antioxidant in *Yoom Noon* rice.

Phytic acid (PA) is widely recognized as an anti-nutritional factor [60]. PA may have potential health benefits such as antioxidant, anti-cancer, and anti-coronary disease properties [61]. High-phytate grain-based foods are widely thought to increase iron and zinc deficiency in developing countries, whereas excreted phytate contributes to environmental pollution by being washed into surface water in developed countries where livestock are largely given grain-based feed [62]. Table 2 shows the PA content of three different processed *Yoom Noon* rice. The highest PA content was found in brown rice (11.2 mg/100 g), which was 1.17 times greater than that of germinated brown rice (*p* < 0.05). Bagchi et al. [60] proposed that endogenous phytase activity is primarily responsible for PA retention in rice. Endogenous phytase activity and PA diffusion into the soaking medium are two factors that could explain why germinated brown rice retains less PA than brown rice. Perlas and Gibson [63] suggested that endogenous phytase is a key factor in lowering PA levels in various rice-based products. Since phytates play a role in germination and are related to the synthesis of gibberellins and abscisic acid, the hormones involved in seed germination [64] lead to the destruction of PA with a significant reduction in PA in germinated *Yoom Noon* rice (Table 2). This result was in agreement with Azeke et al. [64], who observed a significant decrease in PA during rice germination accompanied by increased phytase activity. No PA was detected in white *Yoom Noon* rice (Table 2). PA accumulates primarily in the aleurone layers of rice endosperm, which are distinct from the inner starchy endosperm of rice seeds [65]. Thus, rice polishing may be associated with the absence of PA in white *Yoom Noon* rice. This result was the basis to explain the lower mineral content of white *Yoom Noon* rice (Table 1), as most minerals in rice are associated with phytate [66].

### 3.4. RAG and SAG

The digestibility of starch corresponds directly to the health impact of rice-consuming communities [67]. Generally, in vitro testing can be used to evaluate carbohydrate breakdown and the consequent release of glucose (available for absorption). Englyst et al. [68] established an in vitro approach to dietary carbohydrate digestion and coined two terminologies related to glucose release from carbohydrates. Slowly available glucose (SAG) and rapidly available glucose (RAG) were the terminology used. The amount of glucose rendered available for absorption from a diet during the first 20 min of in vitro incubation is defined as RAG. It essentially refers to glucose liberated from readily digestible starch (RDS) as well as glucose liberated from food in the form of glucose monomer or glucose derived from sucrose. SAG is described as the quantity of glucose released for absorption from food during the incubation times of 20 min and 120 min. In effect, it is the glucose liberated from a diet from slowly digested starch (SDS) as well as any additional glucose released from the food in the form of free glucose. These terms, when used together, may help provide information on how food may behave in vivo [69].

Figure 4a,b show the RAG and SAG contents of white rice, brown rice, and germinated brown rice of *Yoom Noon* rice from Southern Thailand. The non-significant RAG with the highest value was found in white (10.25 mg/100 g) and brown rice (10.04 mg/100 g), while germinated brown rice had the lowest RAG value (5.33 mg/100 g) (Figure 4a). In contrast, the highest SAG value was present in germinated brown rice (9.19 mg/100 g), followed by brown rice (3.58 mg/100 g), and white rice (1.61 mg/100 g) (*p* < 0.05, Figure 4b). Because carbohydrates are the primary driver of rice’s glycemic index, a measure of the quantity of glucose accessible for immediately apparent cellular metabolism or storage for later utilization [70], the high RAG levels of white and brown rice may be related to their high carbohydrate content (Figure 2). According to Abubakar et al. [71], the GI of two germinated Malaysian rice (MR) was lower than that of white and brown type counterparts, except for MR 74 rice. The GI value of rice is thus determined by rice variety. Despite the fact that brown rice had a similar amount of RAG value, it had a higher SAG value when compared to white rice (Figure 4a), which may reduce the net GI of that brown rice (Figure 4b). Imam et al. [72] found that white rice produced more fasting plasma glucose (FPG) in normal rats than brown rice and germinated brown rice, demonstrating that white rice has a high GI. The presence of bioactive compounds in brown rice, particularly phenolic compounds or dietary fiber in germinated brown rice, may be associated with lower glycemic and insulin indices when compared to white rice [73]. According to Nyambe-Silavwe et al. [74], dietary polyphenols may inhibit α-amylase activity, lowering starch digestibility.

## 4. Conclusions

For the first time, the nutritional profiles of *Yoom Noon* white rice, brown rice, and germinated brown rice were determined and presented. The results showed that both macro- and micronutrients vary amongst processed rice. Brown rice, both non-germinated and germinated, had higher levels of all nutrients except carbohydrates and amylose. Regardless of processing condition, Mg was the richest mineral in *Yoom Noon* rice, followed by Zn, Fe, K, Mn, Ca, and Cu. Thiamin was the most abundant in all processed rice, followed by pyridoxine and nicotinamide. Germinated brown rice had the highest thiamin and GABA content, while brown rice had the highest TEF and PA content. RAG values were higher in white rice and brown rice than in germinated brown rice. Germinated brown rice, on the other hand, had the highest SAG value, followed by brown rice and white rice, which may be related to the lower GI of brown rice/germinated brown rice.

Scientific research on the nutritional values of Indigenous rice plays a crucial role in informing policy development and advocacy efforts. Evidence-based knowledge about the unique nutritional benefits of Indigenous rice provides a strong foundation for policymakers, nutritionists, and public health professionals to develop guidelines, programs, and initiatives. Such efforts may include integrating Indigenous rice into national dietary guidelines, promoting local production and consumption, supporting small-scale farmers, and advocating for the conservation and sustainable use of Indigenous rice genetic resources.

## Figures and Tables

**Figure 1 foods-12-02952-f001:**
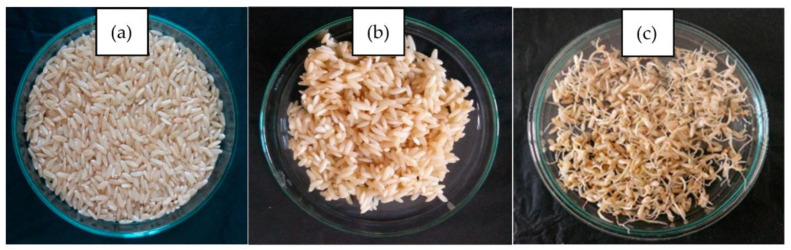
Appearances of white rice (**a**), brown rice (**b**), and germinated brown rice (**c**) of *Yoom Noon* rice from Southern Thailand.

**Figure 2 foods-12-02952-f002:**
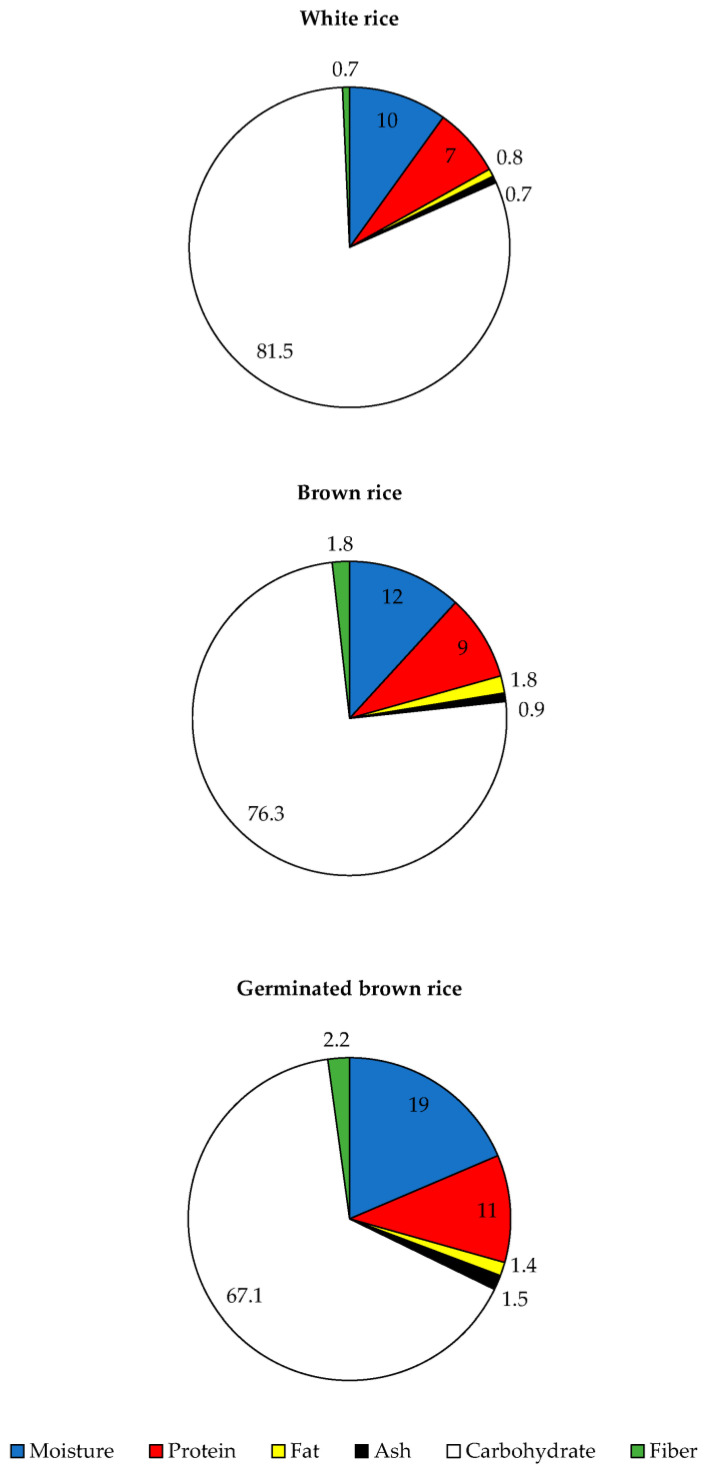
Proximate composition (% wet weight) of white rice, brown rice, and germinated brown rice of *Yoom Noon* rice from Southern Thailand. Values are mean from triplicate determinations.

**Figure 3 foods-12-02952-f003:**
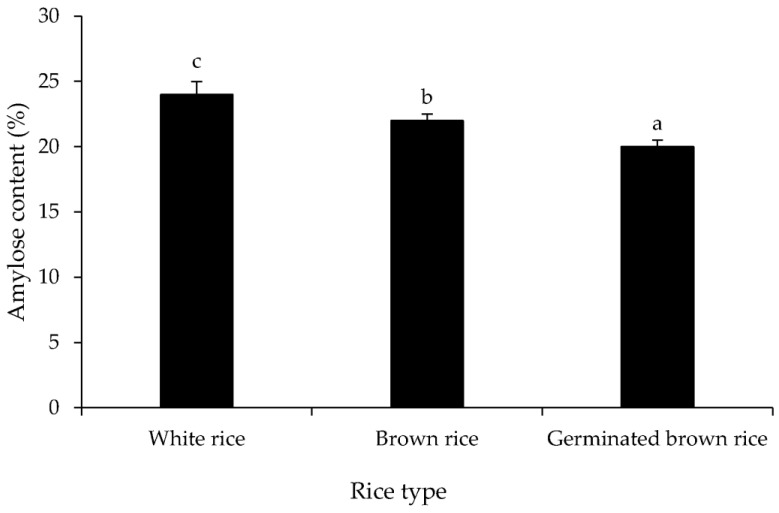
Amylose content of white rice, brown rice, and germinated brown rice of *Yoom Noon* rice from Southern Thailand. Bars represent standard deviation from triplicate determinations. Different letters on the bars indicate significantly different (*p* < 0.05).

**Figure 4 foods-12-02952-f004:**
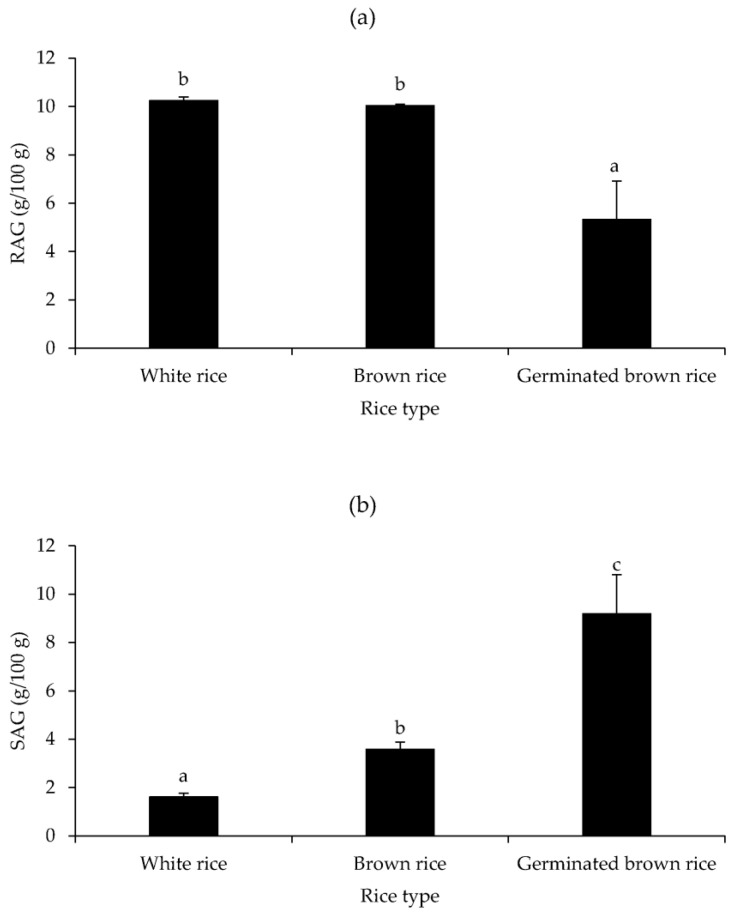
Rapidly available glucose (RAG; (**a**)) and slowly available glucose (SAG; (**b**)) contents of white rice, brown rice, and germinated brown rice of *Yoom Noon* rice from Southern Thailand. Bars represent standard deviation from triplicate determinations. Different letters on the bars indicate significantly different (*p* < 0.05).

**Table 1 foods-12-02952-t001:** Mineral profiles of white rice, brown rice, and germinated brown rice of *Yoom Noon* rice from Southern Thailand.

Mineral (mg/100 g)	White Rice	Brown Rice	Germinated Brown Rice
Magnesium	6.59 ± 0.50 a	10.59 ± 0.71 c	9.00 ± 0.50 b
Zinc	5.98 ± 0.05 b	6.18 ± 0.10 c	4.10 ± 0.10 a
Iron	3.60 ± 0.10 b	4.92 ± 0.12 c	3.45 ± 0.05 a
Potassium	3.40 ± 0.05 b	3.81 ± 0.21 c	2.61 ± 0.11 a
Calcium	1.45 ± 0.40 b	1.66 ± 0.31 c	1.14 ± 0.05 a
Manganese	1.20 ± 0.05 a	2.48 ± 0.10 c	1.75 ± 0.21 b
Copper	0.17 ± 0.02 a	0.23 ± 0.01 b	0.16 ± 0.00 a
Selenium	ND	ND	ND

Values are given as mean ± SD from triplicate determinations. Different letters in the same row indicate significant differences (*p* < 0.05). ND = not detected.

**Table 2 foods-12-02952-t002:** Some vitamins, gamma-aminobutyric acid (GABA), total extractable flavonoid (TEF), and phytic acid contents of white rice, brown rice, and germinated brown rice of *Yoom Noon* rice from Southern Thailand.

Compositions	White Rice	Brown Rice	Germinated Brown Rice
Vitamin			
Thiamin (mg/100 g)	56 ± 2 a	65 ± 4 b	85 ± 4 c
Nicotinamide (μg/100 g)	4 ± 0 a	45 ± 1 d	31 ± 1 b
Pyridoxin (μg/100 g)	18 ± 1 a	44 ± 1 c	21 ± 1 b
Riboflavin	ND	ND	ND
GABA (mg/kg)	ND	197 ± 2 a	585 ± 1 b
TEF (mg RE/100 g)	ND	495 ± 5 b	232 ± 4 a
Phytic acid (mg/100 g)	ND	11.2 ± 0.4 b	9.6 ± 0.5 a

Values are given as mean ± SD from triplicate determinations. Different letters in the same row indicate significant differences (*p* < 0.05). RE = rutin equivalent. ND = not detected.

## Data Availability

The data used to support the findings of this study can be made available by the corresponding author upon request.

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
