# Peer review of "Nutritional Profiles of Yoom Noon Rice from Royal Initiative of Southern Thailand: A Comparison of White Rice, Brown Rice, and Germinated Brown Rice"

_foods, 2023, doi:10.3390/foods12152952_

Round 1
Reviewer 1 Report
The study comparing Yoom Noon Rice variants from the Royal Initiative of Southern Thailand, namely White Rice, Brown Rice, and Germinated Brown Rice, provides valuable insights into their nutritional profiles and potential health benefits.
The research design is commendable, and the results contribute to the existing literature on rice varieties.
However, in order to further strengthen the paper, several areas need to be addressed.
Among the research results of the author, Bioactive components such as TEF and PA of germinated brown rice and Proximate composition analysis results have already been reported to change as rice germinates through many studies in various rice varieties. That is, it can be enough expected that the components of white rice, brown rice, and germinated rice in this sample, Yoom Noon rice, will change similarly to other rice varieties. Therefore, finding the active ingredient or uniqueness of the active ingredient that only Yoom Noon rice or germinated Yoom Noon rice has will be the key.
The purpose of the author's research is to investigate the nutritional benefits of indigenous rice and advance scientific understanding as well as preserve sustainable agriculture and culture. However, to determine what characteristics Yoom Noon rice is different from other rice, or to provide a strong foundation for policymakers, nutritionists, and public health professionals to develop guidelines, effective substances that only Yoom Noon rice has should be presented. There is no novelty in the differences between brown rice and germinated brown rice, Yoom Noon brown rice, and germinated brown rice in their characteristics.
Line67 bioprocessing startegy - bioprocessing strategy
Line100 antioxidantive – antioxidative
Line 131, there is a lack of information about Yoom Noon rice used in the experiment. Although the author stated that it germinated for 96 hours using rice with a germination rate of 90%, this image does not appear to show an equivalent degree of sprouting. It is necessary to check. In addition, there is no explanation for whether the germinated brown rice has the same moisture content as white rice and brown rice when used in the experiment.
With well-written sentences and appropriate terminology use, the author's writing style reveals a good command of the English language. The presented research findings are easier to understand because of the language's clarity.
Author Response
The study comparing Yoom Noon Rice variants from the Royal Initiative of Southern Thailand, namely White Rice, Brown Rice, and Germinated Brown Rice, provides valuable insights into their nutritional profiles and potential health benefits.
The research design is commendable, and the results contribute to the existing literature on rice varieties.
Ans: Thank you very much.
However, in order to further strengthen the paper, several areas need to be addressed.
Among the research results of the author, Bioactive components such as TEF and PA of germinated brown rice and Proximate composition analysis results have already been reported to change as rice germinates through many studies in various rice varieties. That is, it can be enough expected that the components of white rice, brown rice, and germinated rice in this sample, Yoom Noon rice, will change similarly to other rice varieties. Therefore, finding the active ingredient or uniqueness of the active ingredient that only Yoom Noon rice or germinated Yoom Noon rice has will be the key.
Ans: The antioxidant components and antioxidant activity of Yoom Noon in comparison to other indigenous rice have been published. So, in the Introduction, we stated that “Previously, the antioxidant compositions and in vitro antioxidant functionality of Yoom Noon rice processed with different conditions (white rice, brown rice, germinated brown rice, and rice grass) were studied alongside four varieties of Southern Thai rice (Khai Mod Rin, Kaab Dum, Look Lai, and Yar Ko) [6]. Rice's antioxidant compounds and antioxidative capacities were shown to be significantly influenced by both variety and processing. All rice cultivars contained high quantities of total extractable phenolic compounds and carotenoids, particularly rice grass and germinated brown rice [6]. Furthermore, a greater g-oryzanol content was discovered following germination. After sprouting, all rice cultivars had greater phenolic compounds, ascorbic acid, and carotenoid contents. Overall, the highest total phenolic content was found in Yoom Noon rice grass. The greatest g-oryzanol content was found in Yoom Noon's germinated brown rice. The aqueous extracts of all rice cultivars had exceptional ABTS free radical scavenging activity [6]. The findings can be applied as a guideline to select the best rice variety and primary processing method to meet the requirements of farmers who like to advance rice as a functional food ingredient and to encourage health-conscious customers to consume indigenous rice, with Yoom Noon being a good choice as a result of the study.”
However, the nutritional profile of Yoom Noon rice was not mentioned in that study, which is essential from a nutritional standpoint. Therefore, we indicated in the final paragraph of the Introduction that “However, it is important to determine the quality of the nutritional value of Yoom Noon rice as a fundamental standard for healthy consumption in order to consider its potential applications and to support the concept of healthy food ingredients. Therefore, the goal of this study was to examine the nutritional profiles of the white, brown, and germinated forms of the Yoom Noon rice from Southern Thailand. Investigating the nutritional benefits of indigenous rice not only advances scientific understanding but also sets the pathway for supporting sustainable agriculture and cultural preservation.”
The purpose of the author's research is to investigate the nutritional benefits of indigenous rice and advance scientific understanding as well as preserve sustainable agriculture and culture. However, to determine what characteristics Yoom Noon rice is different from other rice, or to provide a strong foundation for policymakers, nutritionists, and public health professionals to develop guidelines, effective substances that only Yoom Noon rice has should be presented. There is no novelty in the differences between brown rice and germinated brown rice, Yoom Noon brown rice, and germinated brown rice in their characteristics.
Ans: Basic information, such as food composition data, is critical for future application in the field of food science. Although nutrition data for rice processing can be predicted, some nutrients can vary based on the rice cultivar and processing circumstances. As a result, the novelty of this study is the description of the nutritional profile of Yoom Noon rice in three different forms, white rice, brown rice, and germinated brown rice, which has not previously been documented. In addition, the RAG and SAG values were calculated for the first time in this rice. All of the data can be used to support the application and citation as nutritional information for food science.
Line67 bioprocessing startegy - bioprocessing strategy
Ans: Done.
Line100 antioxidantive – antioxidative
Ans: Done.
Line 131, there is a lack of information about Yoom Noon rice used in the experiment. Although the author stated that it germinated for 96 hours using rice with a germination rate of 90%, this image does not appear to show an equivalent degree of sprouting. It is necessary to check. In addition, there is no explanation for whether the germinated brown rice has the same moisture content as white rice and brown rice when used in the experiment.
Ans: The information about Yoom Noon rice used in the experiment was updated.
“A native Southern Thai non-glutinous rice (Oryza sativa L.), var. Yoom Noon, was harvested in Ban Phoeng, Pak Phanang, Nakhon Si Thammarat, Thailand. A total of 300 kg of paddy rice was used. White and brown rice were manufactured using a domestic method and nutritional contents were compared to germinated brown rice (Fig. 1). To acquire brown rice samples with intact bran layers and germ, the paddies were milled to remove the hull using a home-scale miller model (THU35B; SATAKE, Hiroshima, Japan). The bran layer was removed during the next round of milling, yielding white rice samples. The rice seeds were tested for germinability (at least 90% germination) before being used to produce germinated brown rice. To prepare the germinated brown rice, brown rice was soaked in water (pH =5) with a brown rice-to-water ratio of 1:4 for 96 h at 35 °C, changing the water every 6 h, and then towel-dried [6]. To prepare the samples for analysis, white rice, brown rice, and germinated brown rice were ground for 5 min using a grinder (MK 5087M Panasonic Food Processor, Selangor Darul Ehsan, Malaysia). The samples were sealed in ethylene-vinyl alcohol copolymer (EVOH) bags and stored at -20 °C until being used. The storage period was no longer than 1 month.”
We sincerely apologize for the error. We used the incorrect image in the first submission. The image has been changed and the degree of sprouting can now be seen in the equivalent way. As can be seen in the data of proximate composition (Fig. 2), the moisture content varied between samples depending on the processing. Section 3.1 provides an explanation of moisture content.
“Moisture is the second most common component found in all processed Yoom Noon rice, with germinated brown rice having the highest moisture content (19%). This was because water was significantly diffused to the rice grain during the immersion step, which was required for germination and sprouting. Brown rice and white rice had moisture contents of 12% and 10%, respectively, which were within the 11-14% range for standard rice moisture content, allowing for long-term storage [27].”

Reviewer 2 Report
Ms. Ref. No.: Foods 2516727
Nutritional Profiles of Yoom Noon Rice from Royal Initiative of Southern Thailand: A Comparison of White Rice, Brown Rice, and Germinated Brown Rice
The objective of the work is to study the nutritional profile of Yoom Noon rice comparing processing and germination. The methodology is generally well defined. And the results are consistent with previous studies. However, in most cases they are predictable.
The manuscript has some minor changes must be considered.
1. In the Materials and Methods section:
In 2.2. White Rice, Brown Rice, Germinated Brown Rice Samples subsection should describe how the processing has been to obtain white rice.
2. In the Conclusions section:
In the conclusions section, the results are described again. The conclusions of the work must be rewritten.
Author Response
Ms. Ref. No.: Foods 2516727
Nutritional Profiles of Yoom Noon Rice from Royal Initiative of Southern Thailand: A Comparison of White Rice, Brown Rice, and Germinated Brown Rice
The objective of the work is to study the nutritional profile of Yoom Noon rice comparing processing and germination. The methodology is generally well defined. And the results are consistent with previous studies. However, in most cases they are predictable.
Ans: Thank you very much. Although nutrition data for rice processing can be predicted, some nutrients can vary based on the rice cultivar and processing circumstances. As a result, the novelty of this study is the description of the nutritional profile of Yoom Noon rice in three different forms, white rice, brown rice, and germinated brown rice, which has not previously been documented. In addition, the RAG and SAG values were calculated for the first time in this rice. All of the data can be used to support the application and citation as nutritional information for food science.
The manuscript has some minor changes must be considered.
- In the Materials and Methods section:
In 2.2. White Rice, Brown Rice, Germinated Brown Rice Samples subsection should describe how the processing has been to obtain white rice.
Ans: The detail was added. “A native Southern Thai non-glutinous rice (Oryza sativa L.), var. Yoom Noon, was harvested in Ban Phoeng, Pak Phanang, Nakhon Si Thammarat, Thailand. A total of 300 kg of paddy rice was used. White and brown rice were manufactured using a domestic method and nutritional contents were compared to germinated brown rice (Fig. 1). To acquire brown rice samples with intact bran layers and germ, the paddies were milled to remove the hull using a home-scale miller model (THU35B; SATAKE, Hiroshima, Japan). The bran layer was removed during the next round of milling, yielding white rice samples.”
- In the Conclusions section:
In the conclusions section, the results are described again. The conclusions of the work must be rewritten.
Ans: The conclusion was restructured. First, based on the results, the comprehensive nutritional profiles of Yoom Noon white rice, brown rice, and germinated brown rice were drawn. Then, we emphasized how our information on the nutritional value of indigenous rice may help drive advocacy and policy-making activities. It also offers a solid framework on which government officials, nutritionists, and public health experts can develop policies, initiatives, and programs. Such efforts may include integrating indigenous rice into national dietary guidelines, promoting local production and consumption, supporting small-scale farmers, and advocating for the conservation and sustainable use of indigenous rice genetic resources.

Reviewer 3 Report
In this manuscript, the authors described Nutritional Profiles of Yoom Noon Rice : A Comparison of White Rice, Brown Rice, and Germinated Brown Rice. The results and findings of this study have the potential to contribute to the long-term food sustainability and security. This manuscript could be considered after the following concerns are addressed by making the required revisions.
If you have found any significant differences between the characteristics of this rice and other rice varieties in this study, please describe them with emphasis.
Could you please show the weight of one thousand or one grain og white rice, brown rice, and germinated brown rice? t would be more comprehensible if the weight of rice grains per wet weight and per dry weight could be compared with and without processing.
Fig and Table
Moisture content differs among white rice, brown rice, and germinated brown rice. When comparing components, it is also necessary to discuss the comparative results in terms of content per dry matter weight in addition to per wet matter weight.  This would change the results of statistical tests.
Fig.2 L239-L244
I can understand the elemental content of germinated vs. brown rice decreasing during soaking in water, but am interested in the reasons for the increase with respect to those that do increase. For example, the protein content is increased in germinated brown rice as compared to brown rice. The protein content is calculated by multiplying the total nitrogen content by factor 5.95. Where does this nitrogen source come from that would be greater without nitrogen in the soaking water? Does it increase by taking up nitrogen from the air during the soaking of the rice grains?
Fig.2, Fig.3, L281-L309
The authors described amylose as low molecular weight and more readily converted to water soluble than amylopectin. Did the amylose content change significantly with processing compared to the starch content?
Author Response
In this manuscript, the authors described Nutritional Profiles of Yoom Noon Rice : A Comparison of White Rice, Brown Rice, and Germinated Brown Rice. The results and findings of this study have the potential to contribute to the long-term food sustainability and security. This manuscript could be considered after the following concerns are addressed by making the required revisions.
If you have found any significant differences between the characteristics of this rice and other rice varieties in this study, please describe them with emphasis.
Ans: The differences between our data and those from other studies can be observed in the examples below.
Similarly, the higher presence of non-starch components in brown rice resulted in a lower amylose content [41]. This was in line with the results of Chatterjee and Das [41], who observed that the amylose content of polished samples ranged from 11.06 to 19.40% in all ten rice varieties, due to the fact that starch is mostly concentrated in the endosperm and less in the bran.
All minerals reported in this study had a higher content than 10 dehulled rice varie-ties from Pakistan [45], indicating that an indigenous Yoom Noon rice is a rich source of minerals. According to Sangha and Sachdeva [46], different rice genotypes have different mineral concentrations. There were some similarities and differences between our study and Huang et al. [47], who observed that brown rice contains an average amount of minerals, with a clear presence of P (4652 mg/kg) and K (3810 mg/kg), followed by Mg (1558 mg/kg), and a very low amount of Zn (34 mg/kg) and Fe (12 mg/kg). The difference between our study and others could be attributed to mineral content accumulation in grains being significantly affected by planting site with variations in soil composition [48].
There was some variation in individual vitamin concentration among different processed Yoom Noon rice, with germinated brown rice having the highest thiamin content by roughly 1.52 and 1.31 folds when compared to white and brown rice, respectively (Table 2). This corresponded to our earlier report [2], which showed that thiamin content increased during germination of some Thai rice cultivars. Similarly, germinated Korean rice had a 1.70-2.10 folds increase in thiamin content when compared to brown rice [54].
The highest GABA content was found in germinated Yoom Noon rice (585 mg/kg), which was 2.97 times greater than brown rice (p < 0.05, Table 2). GABA was not detected in white rice. Chaijan and Panpipat [2] observed that the GABA content of two Thai rice varieties, Khemtong and Khai Mod Rin, increased during germination. Similarly, the GABA content of rough rice increased from 15.34 mg/100 g to 31.79 mg/100 g after germination [30].
Could you please show the weight of one thousand or one grain og white rice, brown rice, and germinated brown rice? t would be more comprehensible if the weight of rice grains per wet weight and per dry weight could be compared with and without processing.
Ans: I appreciate your thoughtful advice very much. Since we no longer have the samples, we apologize for for the inconvenience. In our upcoming investigations, we will take this advice into account. All of the samples were, however, examined for moisture content and reported on a wet weight basis. This data can be used for purposes of comparison.
Fig and Table
Moisture content differs among white rice, brown rice, and germinated brown rice. When comparing components, it is also necessary to discuss the comparative results in terms of content per dry matter weight in addition to per wet matter weight.  This would change the results of statistical tests.
Ans: In order to link the data from this study with reality, we compared data based on wet weight basis. The term "wet weight" refers to a sample with its typical moisture content, which was defined in Figure 1. Additionally, all rice samples were cooked using the same technique to produce cooked rice, which was then tested for RAG and SAG. The RAG and SAG findings were then also presented by means of wet weight. We hope you'll understand our intentions. In the future, comparisons, however, will be made using a dry basis.
Fig.2 L239-L244
I can understand the elemental content of germinated vs. brown rice decreasing during soaking in water, but am interested in the reasons for the increase with respect to those that do increase. For example, the protein content is increased in germinated brown rice as compared to brown rice. The protein content is calculated by multiplying the total nitrogen content by factor 5.95. Where does this nitrogen source come from that would be greater without nitrogen in the soaking water? Does it increase by taking up nitrogen from the air during the soaking of the rice grains?
Ans: We stated that “Protein content ranged from 7 to 11%, while lipid content ranged from 0.8 to 1.8% (Fig. 2). Germinated brown rice had the greatest protein content (11%), followed by brown rice (9%), and white rice (7%). The increase in protein content caused by germination could be attributed to the production of some proteins, peptides, and amino acids during renewed protein biosynthesis [28]. The increased protein content of germinated Yoom Noon brown rice may provide a nutritional advantage.” Although the precise mechanism is uncertain, we mentioned "protein biosynthesis," which occurs when plants mature and germinate.
There was a reference inserted in support. “Protein synthesis is thought to have increased in three stages: (a) concomitant with swelling; (b) during the lag phase between the end of water intake and the start of development; and (c) shortly following protrusion through the seed coat [28].”
de Klerk, G.J.; Smulders, R. Protein synthesis in embryos of dormant and germinating Agrostemma githago L. seeds. Plant Physiol. 1984, 75(4), 929-935.
The assumption was also included. “The additional source of nitrogen that serves as a precursor for protein synthesis may come from the air and water that the seed may absorb during soaking and germination.” To avoid the overstatement, there will be no further extended explanation.
Fig.2, Fig.3, L281-L309
The authors described amylose as low molecular weight and more readily converted to water soluble than amylopectin. Did the amylose content change significantly with processing compared to the starch content?
Ans: From the results we found that “Carbohydrate, primarily starch, was the most abundant macronutrient in all processed Yoom Noon rices, accounting for 67.1 to 81.5% of the total. White rice contained the highest amount of carbohydrate, followed by brown rice and germinated brown rice (p < 0.05).” and “White rice had the highest amylose content, approximately 24%, followed by brown rice (22%), and germinated brown rice (20%), respectively (p < 0.05).”

Round 2
Reviewer 1 Report
Revised manuscript confirmed